# Identification and Cloning of a CC-NBS-NBS-LRR Gene as a Candidate of *Pm40* by Integrated Analysis of Both the Available Transcriptional Data and Published Linkage Mapping

**DOI:** 10.3390/ijms221910239

**Published:** 2021-09-23

**Authors:** Huai Yang, Shengfu Zhong, Chen Chen, Hao Yang, Wei Chen, Feiquan Tan, Min Zhang, Wanquan Chen, Tianheng Ren, Zhi Li, Peigao Luo

**Affiliations:** 1Provincial Key Laboratory for Plant Genetics and Breeding, College of Agronomy, Sichuan Agricultural University, Chengdu 611130, China; yanghuai202103@163.com (H.Y.); zhongshengfu@tom.com (S.Z.); icbrcc2018@163.com (C.C.); 2020101002@stu.sicau.edu.cn (H.Y.); 3-spongefarmer@163.com (W.C.); FeiquanTan_1@163.com (F.T.); renth@sicau.edu.cn (T.R.); lizhi@sicau.edu.cn (Z.L.); 2College of Agronomy & Key Laboratory for Major Crop Diseases, Sichuan Agricultural University, Chengdu 611130, China; yalanmin@126.com; 3State Key Laboratory for Biology of Plant Diseases and Insect Pests, Institute of Plant Protection, Chinese Academy of Agricultural Sciences, Beijing 100193, China; wqchen112@ippcaas.cn

**Keywords:** wheat, powdery mildew, express sequence, variation annotation, linkage region

## Abstract

Wheat powdery mildew, caused by the obligate parasite *Blumeria graminis* f. sp. *tritici*, severely reduces wheat yields. Identifying durable and effective genes against wheat powdery mildew and further transferring them into wheat cultivars is important for finally controlling this disease in wheat production. *Pm40* has been widely used in wheat breeding programs in Southwest China due to the spectrum and potentially durable resistance to powdery mildew. In the present study, a resistance test demonstrated that *Pm40* is still effective against the *Bgt* race E20. We identified and cloned the *TraesCS7B01G164000* with a total length of 4883 bp, including three exons and two introns, and encoded a protein carrying the CC-NBS-NBS-LRR domain in the *Pm40*-linked region flanked by two EST markers, *BF478514* and *BF291338*, by integrating analysis of gene annotation in wheat reference genome and both sequence and expression difference in available transcriptome data. Two missense mutations were detected at positions 68 and 83 in the CC domain. The results of both cosegregation linkage analysis and qRT-PCR also suggested that *TraesCS7B01G164000* was a potential candidate gene of *Pm40*. This study allowed us to move toward the final successfully clone and apply *Pm40* in wheat resistance improvement by gene engineering.

## 1. Introduction

Powdery mildew, caused by *Blumeria graminis* f. sp. *tritici* (*Bgt*), is a widespread fungal disease that leads to a serious decrease in wheat production worldwide [1,2,3]. In Southwest China, powdery mildew is becoming increasingly devastating and could soon be the most damaging disease in wheat production because the conditions favor pathogen growth [4], increased nitrogen fertilizer and irrigation input, and this disease receives less attention compared to that of stripe rust caused by *Puccinia striiformis* f. sp. *tritici* [5]. Chemical control measures are widely used in various regions, but developing and applying resistant cultivars are the more effective, economical, and environmentally friendly means of controlling this disease [6,7]. Therefore, improving cultivar resistance by identifying and transferring broad spectrum and durable resistance *Pm* genes in wheat breeding programs is urgently needed.

Various scientists have been committed to discovering new loci or alleles conferring resistance to powdery mildew, and, to date, 99 *Pm* genes at 60 loci (*Pm1–Pm68*) that confer resistance to powdery mildew have been publicly reported [7,8,9,10]; these genes have been assigned to almost all chromosomes except 3D and 4D. Unfortunately, only a few *Pm* genes, such as *Pm21,* have been successfully used in the development of resistant cultivars because of the loss of resistance resulted from the rapid variation in *Bgt* races [11], linkage drag [12,13], and the shortage of information on both the sequence and function of *Pm* genes [14].

In the past, some *Pm* genes, such as *Pm2*, *Pm5*, *Pm6*, and *Pm8*, have been successfully applied in resistant wheat breeding programs in China [15]. At present, although *Pm21* is still effective and the most widely applied *Pm* gene in the country [16], some newly emerged and virulent isolates have been reported to overcome *Pm21* resistance in various regions [17,18]. We found that *Pm40* still confers strong resistance to powdery mildew in various regions [2,19,20]. It was also effective against nearly all isolates collected from the main Chinese wheat growing regions, which indicated that *Pm40* could have a large potential role in the improvement in resistance against powdery mildew in China in the future, especially in the post-*Pm21* era [7]. Therefore, identifying and cloning the candidate gene of *Pm40* is a valuable endeavor.

Due to its large genome, fragmented genomic information, and high proportion of repetitive sequences, it is difficult to clone genes from hexaploid wheat. To date, among the known *Pm* genes, only *Pm1* [21], *Pm2* [22], *Pm3* [23], *Pm5* [24], *Pm8* [25], *Pm21* [26], *Pm24* [27], *Pm38/Lr34/Yr18/Sr57* [28], *Pm41* [29], *Pm46/Yr46/Lr67/Sr55* [30], and *Pm60* [31] have been cloned. Fast progress in high-quality assembly and annotation of the Chinese Spring wheat reference genome (IWGSC RefSeq v1.0) has provided available information and resources to identify candidate genes in a given chromosomal region by gene annotation [32]. In addition, various available transcriptome expression sequences during the interaction between the pathogen and the host have also been helpful for the identification of candidate resistance genes [33,34,35]. Both cosegregation and qRT-PCR analyses are common and useful methods to validate the function of candidate genes [4,36].

Some studies have shown that these cloned resistance genes from different species usually have similar structural domains, such as Coiled-coil (CC), Toll and interleukin-1 receptors (TIR), nucleotide binding sites (NBS), leucine-rich repeats (NLR), and receptor-like proteins/kinases (RLPs/RLKs) [15,37]. Sequence analysis further found that most of the reported powdery mildew resistance genes in wheat usually carry CC-NBS-LRR (CNL) or RLK domains [4,29]. Therefore, the genes carrying CNL and RLK domains should have a higher consideration during the process of screening and identifying candidate genes for powdery mildew resistance in wheat.

In 2007, we first identified *PmE* controlling resistance to powdery mildew in the wheat line YU25 by genetic analysis [38], which was consequently mapped to chromosome 7B and formally named *Pm40* [19]. In a later study, *Pm40* was further mapped to a short chromosomal region of 7BS flanked by *Xwmc335* (with distances of 0.58 cm) and *BF291338* (with distances of 0.26 cm) [20]. With the release of the Chinese spring reference genome sequence, it was feasible and workable to identify the effective candidate gene of *Pm40* because the high-quality reference genome information was available.

We knew through experience that it was difficult to further narrow the chromosomal region, which was possibly related to the origin of *Pm40*. Originally we thought *Pm40* was derived from *Th. intermedium* because it had Line Yu25 conferring *Pm40* resistance in its pedigree [19]. Subsequent evidence, including the uniformity of agronomic traits and the good genetic stability of powdery mildew resistance in many years [2,7], the genetic behavior as a normal Mendelian factor [19,20,38], the wheat-specific amplicons produced by polymerase chain reaction (PCR) amplification, and the similar marker order with the similar total length of the linkage mapping of *Pm40* [19,20], supported the view that *Pm40* may result from DNA sequence change or chromosomal rearrangement of wheat self-genome during the wild cross process rather than direct transfer from *Th. intermedium.* Recently, many studies clearly found that the wide cross between wheat and rye (*Secale cereal*) caused wheat DNA sequence change or chromosomal rearrangement [39,40], and a similar result was also found when the wide cross between *Arabidopsis thaliana* and *A. lyrata* was executed [41]. In addition, a copper-binding protein gene (*WCBP1*) produced from the wheat genomic DNA change was possibly involved in the resistance to wheat stripe rust in YU25 [42]. Therefore, these results indirectly reinforce the view that *Pm40* could be derived from the wheat-self genome changes.

In addition, regarding the *Pm40* resistance mechanism to powdery mildew, some physiological and biochemical characteristics were also compared between L693 carrying *Pm40* and L1034 without *Pm40* [2]. Recently, transcriptome analysis showed that ATPase, PSEPS, and the HSPs of *Bgt* possibly played the crucial roles in establishing the pathogenesis of compatible interactions in hosts without *Pm40* [6]. In contrast, in wheat line carrying *Pm40*, the decline in photosynthesis caused by the downregulation of photosynthesis-related genes and the subsequent accumulation of H_2_O_2_ could be an important signal mechanism of incompatible pathogenesis [43]. The analysis did not directly identify the candidate genes of *Pm40* [6,43], but the available expression sequences are very important for identifying those candidate genes.

In this study, we had the following objectives: to further test the resistance of *Pm40* by inoculation with *Bgt* race E20; to identify candidate resistance genes by comprehensive analysis using gene annotation data from the IWGSC RefSeq v1.0 in the physical region flanked by two EST markers, *BF478514* and *BF291338* [20], on chromosome 7B and both sequence and expression differences in the available transcriptome data; to clone the *TraesCS7B01G164000* sequence and further confirm the sequence differences between L658 carrying *Pm40* and L958 lacking *Pm40*; and to validate the candidate gene *TraesCS7B01G164000* by cosegregation and qRT-PCR analyses. Comprehensively, all results supported *TraesCS7B01G164000* as a potential candidate gene of *Pm40*.

## 2. Results

### 2.1. Identification of Pm40 Effectiveness against Bgt Infection

The wheat line L658 and its resistant parent YU25 exhibited high resistance (with IT = 0) to powdery mildew, while line L958 and its susceptible parent cultivar MY11 were susceptible (Figure 1A). Cytological observations further found a significant hypersensitive response in the leaf cells of L658 at 72 hpi, while this phenomenon was not observed in L958 leaf cells (Figure 1B). The results suggested that *Pm40* was effective against the E20 race of *Bgt*.

### 2.2. Functional Annotation of the Genes within the Region Flanked by BF478514 and BF291338

A total of 185 genes were annotated in the 37 Mb physical region flanked by *BF478514* and *BF291338* on chromosome 7BS, among which only one CNL gene and seven RLK genes were putative functional candidate genes (Table 1). The eight genes were clustered into two narrow subregions: one included five genes close to marker *BF478514*, and the other included three genes close to marker *BF291338* (Figure 2A). Based on the genetic distance and expression profile, we paid more attention to the three genes (*TraesCS7B01G162500* and *TraesCS7B01G164600* encoding RLK proteins, and *TraesCS7B01G164000* encoding a CNL protein) because the cluster was close to the marker *BF291338* and the genetic distance was closer to *Pm40*. Furthermore, the EST marker *BE446359,* with a genetic distance of 0.28 cm from *Pm40* [20], coincided with *TraesCS7B01G162500*, so *TraesCS7B01G162500* was ruled out as a candidate gene of *Pm40*. Therefore, we focused on the two genes: *TraesCS7B01G164000* encoding a CNL protein and *TraesCS7B01G164600* encoding an RLK protein could be candidate genes of *Pm40.*

### 2.3. Polymorphic EST Sequences in the Pm40 Region

Transcriptomic data showed that 100 out of 185 genes in the *Pm40* region expressed after inoculation with *Bgt*. The 100 expressed genes harbor 458 high-quality variants mainly close to the marker *BF291338* (Figure 2B). SnpEff analysis discovered that the 458 variants resulted in 768 genetic effects, including 19 high-impact, 55 moderate-impact, 627 modifier-impact, and 67 low-impact effects. Among these effects, 19 high-impact effects were contained in nine genes, and 55 moderate-impact effects were contained in 25 genes; a total of 34 genes harbored high or moderate-impact effects. Interestingly, 24 out of the 34 genes were clustered in an approximately 8 Mb region close to *BF291338* (Figure 2C). In addition, among the 34 genes, functional analysis only identified *TraesCS7B01G162500* and *TraesCS7B01G164000* as typical plant disease resistance genes (Table 1), and three variants in *TraesCS7B01G162500* and two variants in *TraesCS7B01G164000* showed moderate-impact effects (Table 2).

Frameshift: A sequence variant that causes a disruption of the translational reading frame; Splice acceptor: A splice variant that changes the 2 base pair region at the 3’ end of an intron; Splice donor: A splice variant that changes the 2 base pair region at the 5’ end of an intron; Stop gained: A sequence variant whereby at least one base of a codon is changed, resulting in a premature stop codon; Missense variant: A sequence variant that changes one or more bases, resulting in a different amino acid sequence but where the length is preserved. The influence of variation on genes was classified into Variants_impact_HIGH and Variants_impact_MODERATE, proposed by SnpEff.

### 2.4. Differential Expression Level of Genes within Pm40 Candidate Region

Only 24 out of the 100 expressed genes within the *Pm40* mapped region were differentially expressed between the resistant line L658 and susceptible sister line L958 after inoculation with *Bgt* (Figure 3A), and five putative functional candidate genes screened out in the above analysis were detected in L658 and L958 (Figure 3A). These DEGs were mainly involved in lipid transport and metabolism, structural ribosome constituents, DNA binding, sodium ion transport, microtubule binding, RNA methylation, and unknown functions (Appendix A); they exhibited similar expression trends in the two lines (Figure 3B). Nearly half of the DEGs were also present in the 8 Mb region close to *BF291338* (Figure 3A). Moreover, most of the DEGs in this 8 Mb region also had variants that probably affected genes (Figure 3A).

### 2.5. Screening and Cloning of TraesCS7B01G164000 and TraesCS7B01G164600

A 4883 bp sequence covering the whole coding region of *TraesCS7B01G164000* was obtained from the wheat lines L658 and L958 by PCR product sequencing; this sequence included three exons and two introns and encoded a CNL protein carrying two adjacent repeated NBS domains (Figure 4A). Two SNPs were found by cloning sequence alignment between L658 and L958 (Figure 4B), and both SNPs occurred in the region encoding the CC domain. These SNPs resulted in a transition of Met to Val at amino acid 68 and Glu to Lys at amino acid 83 in L658 compared with L958 (Figure 4B). The prediction of secondary structure showed that the mutation of 83th might cause the change in the secondary structure of the amino acid sequence (Appendix A), and the missense mutation might further cause the function of *TraesCS7B01G164000* to be deleterious by PROVEAN analysis (Appendix A).

We also obtained a 1882 bp sequence covering the whole coding region of *TraesCS7B01G164600* by sequencing PCR products amplified in wheat lines L658 and L958. Sequence alignment showed that the *TraesCS7B01G164600* sequences were identical between wheat lines L658 and L958. Therefore, *TraesCS7B01G164000* was the only candidate gene due to no difference in expression and sequence of *TraesCS7B01G164600* in L658 and L958 after inoculation with *Bgt*.

### 2.6. Validation of the Candidate Gene by Cosegregation with Powdery Mildew Resistance

Two AS-PCR markers were developed based on the SNPs within *TraesCS7B01G164000*: one marker for the A/G single nucleotide variant responsible for the transition from Met to Val was effectively genotyped, while the other marker for the G/A single nucleotide variant did not amplify clear bands. Cosegregation analysis showed that all 39 resistant lines produced PCR products matching those from L658, while all 30 susceptible lines produced products matching those from L958 (Figure 4C), which indicated that the candidate gene cosegregated with powdery mildew resistance.

### 2.7. Expression Analysis of TraesCS7B01G164000 after Bgt Inoculation by qRT-PCR

One set of qRT-PCR primers was designed for gene expression analysis according to the sequence of the third exon of *TraesCS7B01G164000* (Appendix A). *TraesCS7B01G164000* exhibited obvious differential expression at different stages of inoculation with *Bgt* (Figure 5). First, the expression level of *TraesCS7B01G164000* in L658 was significantly higher than that in L958 at 0 hpi without inoculation. Second, the highest expression level in L658 occurred at 6 hpi, while that in L958 occurred at 24 hpi; interestingly, the lowest expression level in L658 was detected at 12 hpi, while that in L958 occurred at 48 hpi. Finally, the trend for the expression level changes of *TraesCS7B01G164000* from 48 hpi to 72 hpi was similar between L658 and L958.

## 3. Discussion

Identification and cloning of the candidate *Pm40* gene are valuable for breeding applications regarding wheat resistance improvement as well as for elucidating the molecular mechanism that wheat resists the infection of the pathogen *Bgt*. However, limited and fragmented information on the wheat genome has resulted in little progress in wheat resistance gene cloning in the past [44]. Although transcriptome analysis has been indicated to be an effective strategy for identifying candidate genes, some important candidate genes were possibly omitted due to the unfit analysis idea of transcriptomic data and the identification of too many candidate genes [45], which made cloning candidate genes difficult. In the present study, we comprehensively used the available gene annotation data in the high-quality assembled wheat genome, the previously published linkage genetic map of *Pm40* [19,20] and the recently reported transcriptomic data of both L658 carrying *Pm40* and L958 lacking *Pm40* to identify the candidate gene of *Pm40* [43].

### 3.1. The Broad Spectrum and Putatively Durable Resistance of Pm40

The powdery mildew resistance gene *Pm40* was first reported in 2007 and tentatively named *PmE*, and it was found to confer strong resistance to powdery mildew in wheat [38]. The *Pm40* has been known to confer strong resistance to toxic and prevalent *Bgt* 15 for more than ten years in Southwest China [2,6,19,20]. Moreover, the wheat line L658 carrying *Pm40* was also resistant to all 28 *Bgt* isolates collected from various regions of China [7]. In the present research, the resistance assessment indicated that *Pm40* still provides effective resistance to the currently prevalent *Bgt* race E20 in Wenjiang, Sichuan Province. Moreover, the *Pm40* gene has been widely used in wheat breeding for many years, and to date, we have not found a loss of resistance from *Pm40* in the field of Southwest China, in which *Bgt* rapidly accumulates variations [46]. Both the persistence and large resistance spectrum of *Pm40* indicated that it could be a durable resistance gene against powdery mildew in wheat.

### 3.2. The Possible Chromosomal Regions Containing the Candidate Pm40 Gene

Two similar high-density genetic linkage maps of *Pm40* with total lengths of 6.18 and 6.38 cm were produced by using 579 individuals from the F_2:3_ population derived from the cross L693/L1034 and 3420 individuals from the F_2:3_ population derived from the cross L658/L958; in both maps, *Pm40* was flanked by *Xwmc335* and *BF291338* [7,20].

It is well known that some SSRs, especially those in intergenic regions, cannot be accurately assigned to the physical map [47]; therefore, the EST marker *BF478514* next to *Xwmc335* was treated as the corresponding flanking marker on the side of *Xwmc335*. The physical interval between the two markers *BF478514* and *BF291338* in the Chinese Spring wheat reference genome was approximately 37 Mb, in which we identified 185 genes.

One CNL and seven RLK genes were identified from the 185 genes based on functional annotation (Table 1), and their distribution formed two obvious clusters. One cluster, including five genes, was on the side of *BF478514,* and the other, including three genes, was on the side of *BF291338* (Figure 2A). We further focused on the region close to *BF291338* because of the smaller genetic distance between *Pm40* and *BF291338* on the linkage map, and more of both the variants (Figure 2B) and differentially expressed genes (Figure 3A) were found in the region.

Further analysis showed that three out of the five disease resistance genes in the cluster on the side of *BF478514* were not expressed, while there were also no variants in the other two expressed genes. In contrast, all three genes in the cluster on the side of *BF291338* were expressed to various degrees, and there were detectable variants in the two genes (Table 2). Together, the candidate genes of *Pm40* could be in the chromosomal region close to the marker *BF291338*.

### 3.3. TraesCS7B01G164000 as the Putative Candidate Gene of Pm40

In wheat, these cloned resistance genes were mainly classified into CNL- and RLK-encoding genes [4,24,29]. Among the three plant disease resistance candidate genes in the chromosomal region on the side of *BF291338*, *TraesCS7B01G164000* encodes a CNL protein carrying two NBS domains (Figure 4A), while the other two candidates, *TraesCS7B01G162500* and *TraesCS7B01G164600*, are RLK genes (Table 1).

Sequence variation analysis did not detect any variants in the expressed sequence of *TraesCS7B01G164600* between L658 and L958 (Figure 2C). In addition, PCR product sequencing also demonstrated that the sequence of *TraesCS7B01G164600* in L658 was the same as that in L958. Therefore, *TraesCS7B01G164600* should be excluded from the candidate genes of *Pm40*. Although expression of *TraesCS7B01G162500* was observed during pathogenesis after *Bgt* inoculation, and there were also three variants in *TraesCS7B01G162500* (Table 2), we ruled out the possibility that this gene was a candidate gene of *Pm40* because the marker *BE446359,* as one part of *TraesCS7B01G162500,* exhibited incomplete linkage with *Pm40* in two previously reported high-density genetic maps [20].

The *TraesCS7B01G164000* was expressed after *Bgt* inoculation [43]. Additionally, two SNPs were found in *TraesCS7B01G164000* between L658 and L958 (Table 2), and silicon analysis further indicated that both SNPs could result in the change of the amino acid sequence (Appendix A). In addition, the two SNPs were also demonstrated by PCR product sequencing (Figure 4B). Comprehensively, it is reasonable that *TraesCS7B01G164000* is the putative candidate gene of *Pm40*.

### 3.4. Validation of TraesCS7B01G164000 as the Candidate Gene of Pm40

Most R genes that mediate effector-triggered immunity (ETI) associated with HR encode NLR proteins [48,49,50]. To date, almost all cloned *Pm* genes in wheat belong to CNL types, suggesting that CNL proteins play major roles in the innate immune defense of wheat against powdery mildew [29]. In the present study, the leaf cells of L658 carrying *Pm40* showed HR against powdery mildew, and the gene *TraesCS7B01G164000* encoded a CNL-type protein, which indicated that it was possibly the candidate gene of *Pm40*.

A previous study showed that *TraesCS7B01G164000* had different expression trends in L658 and L958 after inoculation with *Bgt*, although it was not a DEG [43]. In fact, the expression of the same gene exhibited slight differences in both expression level and trend in different races [51], and different rank position of this gene in cascade signaling pathway [52]. Therefore, the proportion of genes low abundance expression cannot be directly compared because of both low sequencing depth and stage-specific expression [53,54]. In this study, qRT-PCR analysis, which is usually treated as a powerful tool for the study of low abundance expression [55], shows that the expression of *TraesCS7B01G164000* in L658 cells was upregulated after inoculation with *Bgt* and expression peaked at 6 hpi, which agreed with previous work showed that the genes expression influencing the infection of *Bgt* could occur within 12 hpi [6]. Moreover, most of the DEGs encoding pathogen-related 1 (PR1) and pathogen-related 5 (PR5) were expressed at greater levels in L658 at 24 and 48 hpi, following the peak expression time of *TraesCS7B01G164000*, indicating that *TraesCS7B01G164000* may provide the defense response against wheat powdery mildew.

In this study, missense mutations resulting in amino acid changes located at the 68th and 83th positions in the CC domain of *TraesCS7B01G164000* were identified. The amino acids at these two positions have great differences in physical properties (Appendix A), which may lead to changes in the secondary structure of the CC domain and further cause the function of *TraesCS7B01G164000* to be damaged (Appendix A), as determined by silicon analysis. The molecular mechanism of the CC domain combined with an NBS-LRR domain in mediating plant disease resistance is still controversial, but the CC domain surely plays a key role in the plant resistance response, and some CC domains can independently induce HR [31,56,57,58]. Therefore, the loss of resistance in the sister line L958 could result from two missense mutations in the CC domain. It is generally believed that the NBS-ARC domain can activate the conformation of the NBS-LRR protein and hydrolyze ATP, and studies have shown that the NBS-ARC domain can participate in the recognition of pathogenic effector proteins [59,60]. By domain analysis of the coding sequence of the *TraesCS7B01G164000* gene, it was found that this gene has a special structure different from the typical CNL gene. *TraesCS7B01G164000* contains two NBS domains in the middle of the CC domain and leucine-rich repeats. This unique gene structure could be associated with broad-spectrum and durable powdery mildew resistance.

Allele-specific polymerase chain reaction (AS-PCR) is an effective method for the identification of SNPs based on PCR amplification [61]. We detected an A/T mutation that caused a change in amino acid 68th by AS-PCR in the high-generation homozygous sister line of L658. This result implies that *TraesCS7B01G164000* cosegregated with powdery mildew resistance despite the limited sample size of sister wheat lines.

Comprehensively, the resistance test showed that *Pm40* was effective against the current prevalent *Bgt* race E20. *TraesCS7B01G164000,* encoding a CC-NBS-NBS-LRR protein, was identified as a potential candidate gene of *Pm40* by integrating analysis of the genetic map of *Pm40*, gene annotation and transcriptome data, which was demonstrated as the valid candidate gene of *Pm40* by both cosegregation analysis and qRT-PCR confirmation. *Pm40* has great application potential in wheat breeding, while the AS-PCR cosegregation marker developed with *Pm40* will accelerate the application of *Pm40* in molecular breeding. In addition, cloning *Pm40* is the objective of our next study.

## 4. Materials and Methods

### 4.1. Plant Materials, the Resistance Evaluations, and the Cytological Observations of Cell Death

Wheat line L658 carrying *Pm40* and the susceptible sister line L958 derived from the F_7_ populations of a cross between the susceptible line MY11 and the resistant line YU25 [62] were employed for qRT-PCR analysis. The 69 homozygous inbred lines selected from the F_10_ populations derived from the cross YU25/MY11 were employed for cosegregation analysis. The race E20 of *Bgt* was used for resistance evaluations and was isolated from a mixture collected in Wen Jiang, Chengdu, Sichuan Province (latitude N 30°40′ and longitude E 103°51′) in 2020. Wheat seedlings at the three-leaf stage were inoculated by dusting conidia from sporulating seedlings. The infection types were classified using a rating scale of 0 to 4 approximately 14 days after inoculation [63]. The leaves harvested at 72 h post-inoculation (hpi) were stained by the trypan blue (TPB) staining method [64], and the leaf cell death induced by the HR response was observed and photographed using bright-field microscopy (Nikon ECLIPSE80i, Nikon Corporation, Tokyo, Japan).

### 4.2. Annotation of the Genes within the Physical Region of Pm40

Three molecular markers—*BF478514*, *Xwmc335*, and *BF291338*—were genetically linked with the *Pm40* gene with genetic distance of 1.46 cm, 0.58 cm, and 0.26 cm, and *Pm40* in the middle of *Xwmc335* and *BF291338*, respectively, and both *BF478154* and *Xwmc335* were placed on the same side of the chromosome as *Pm40* [20]. Although *Xwmc335* is closer to *Pm40* than *BF478514* [20], we used *BF478514* as the corresponding flanking marker when the chromosomal region of *Pm40* was concerned because the SSR marker of *Xwmc335* was not accurately mapped to the reference genome of wheat. The physical positions of these markers were determined through the Triticeae Multi-omics Center (http://202.194.139.32/PrimerServer/, accessed on 8 October 2020). Therefore, the physical interval length between the two markers *BF478514* and *BF291338* in the Chinese Spring reference genome was approximately 37 Mb.

### 4.3. Acquisition of RNA Sequencing Data and Detection of Deferentially Expressed Gene

RNA sequencing data (accession number SRP117269) of sister wheat lines carrying and lacking *Pm40* after inoculation with *Bgt* were downloaded from the NCBI Sequence Read Archive (SRA) [6]. Reads obtained by RNA sequencing were then mapped to the IWGSC RefSeq v1.0 Wheat Genome Reference (https://wheat-urgi.versailles.inra.fr/, accessed on 9 October 2020) by HISAT2 [65]. To determine the gene expression level, the normalized expression level FPKM was calculated by StringTie [66]. To filter low-expression genes, the following criteria were used: expression level of genes CPM (count per million) value > 1 and sum of CPM > 3 in all samples. The paired comparison test of EdgeR was employed to detect deferentially expressed genes (DEGs) with the threshold of a false discovery rate (FDR) 0.001 or less and |log2 (a fold change)| ≥ 2 or higher [67]. Transcribed genes were annotated using the Non-Redundant Database, Swiss-Prot Protein Database, and Pfam database.

### 4.4. Variation Extraction, Filtering and Annotation

The variations were detected by comparing the expression sequences of the sister lines carrying and lacking *Pm40* from the original transcriptome sequencing files. SamTools, mpileup, and bcftools were used to extract the variants located on the physical interval between the two markers *BF478514* and *BF291338*, and then they were outputted in VCF format. The filtration conditions were set as follows: Qual ≥ 30; DP ≥ 10. Finally, combined with the gene annotation data from the Wheat Chinese Spring Reference Genome 1.0, SnpEff software was used to annotate and analyze the obtained high-quality variations [68]. The genetic effects of the variations are divided into the following four types: high-impact (e.g., nonsense mutations and frameshift mutations), moderate-impact (e.g., missense mutations), modifier (e.g., intron and intergenic mutations), and low-impact mutations (e.g., synonymous mutations) (see http://snpeff.sourceforge.net for details, accessed on 15 October 2020).

### 4.5. Gene Cloning and Sequence Analysis

The Chinese Spring wheat reference genome sequence was used as a template, and specific primers (Appendix A) were designed to amplify the *TraesCS7B01G164000* and *TraesCS7B01G164600* sequences in the resistant wheat line L658 and susceptible sister line L958. Phanta Max Super-Fidelity DNA Polymerase (Vazyme, Chengdu, China) was used to amplify the *TraesCS7B01G164000* and *TraesCS7B01G164600* sequences. PCR was performed in a PTC-200 thermal cycler (MJ Research, Watertown, MA, USA) under the following conditions: 3 min at 95 °C for predenaturation; 34 cycles of 15 s at 95 °C, 15 s at the annealing temperature (Appendix A), and 2 min 30 s at 72 °C; and a final step of 5 min at 72 °C. The amplified *TraesCS7B01G164000* and *TraesCS7B01G164600* gene fragments were attached to the pClone007 Versatile simple vector (TSINGKE, Chengdu, China) for sequencing at Tsingke (Chengdu, China). After sequencing and splicing, DNAMAN (version 9) software was used to detect nucleotide variation. The functional module of CD—Search in NCBI (https://www.ncbi.nlm.nih.gov/Structure/cdd/wrpsb.cgi, accessed on 17 December 2020) was used to analyze the domain structures of the cloned genes. The online server PROVEAN was used to predict the effect of gene function by amino acid variation (http://provean.jcvi.org/, accessed on 5 March 2021). The cloned genes’ sequences were submitted to NCBI and the assigned GenBank Accession numbers were MZ779114 and MZ779115.

### 4.6. SNP Genotyping by AS-PCR

In AS-PCR, Taq polymerase can effectively carry out DNA synthesis reactions only when the base at the 3′ end of the primer is correctly matched with the template DNA [61,69]. The primers were designed based on the SNP of *TraesCS7B01G164000* detected in wheat L658 and L958, as shown in Appendix A. The specificity of the primers was tested in the IWGSC RefSeq v1.0. PCR was performed in a PTC-200 thermal cycler (MJ Research, Watertown, MA, USA) under the following conditions: 4 min at 94 °C for predenaturation; 32 cycles of 45 s at 94 °C, 45 s at the annealing temperature (Appendix A), and 45 s at 72 °C; and a final step of 7 min at 72 °C. After amplification, a 1.5% agarose gel was used to detect the target bands by agarose gel electrophoresis.

### 4.7. Gene Expression Analysis by qRT-PCR

The seedlings of L658 and L958 were cultured in a light culture chamber under 14 h light/10 h dark conditions at 18 °C and 80% humidity (Microclima MC1750E, Snijders Scientific, Tilburg, Holland). Ten-day-old seedlings of L658 and L958 were inoculated with *Bgt* at densities of approximately 100–200 conidia per mm^2^. The first leaf was randomly clipped from wheat lines L658 and L958 at 0, 6, 12, 24, 48, and 72 hpi. To minimize experimental errors, we performed three biological replicates. Total RNA was extracted from 36 leaf samples using the TRIzol reagent method (Invitrogen Life Technologies, Carlsbad, CA, USA) and repeated 3 times. The integrity and purity of the extracted RNA were measured using an Agilent 2100 BioAnalyzer (Agilent Technologies, Santa Clara, CA, USA) and a Nanodrop ND-1000 spectrophotometer (Thermo Scientific, Wilmington, DE, USA).

The specific primers for qRT-PCR analysis were designed in NCBI Primer-BLAST according to the *TraesCS7B01G164000* exon sequence (Appendix A). qRT-PCR was performed using a CFX96 Real-Time System (Bio-Rad Laboratories, Hercules, CA, USA) with SYBR Green qRT-PCR Master Mix (Omega, Beijing, China). The qRT-PCR execution procedure was as follows: 40 cycles of 30 s at 94 °C, 5 s at 94 °C and 30 s at 60 °C. The melting curve was set at 60 °C to 95 °C with a 0.5 °C increase per step. Then, the 2^−ΔΔCt^ method was used to calculate the relative expression of *TraesCS7B01G164000* [70], with tubulin [71] used as a reference gene. The expression levels of the target genes were analyzed using three biological replicates, in which each contained three technical replicates.

## Figures and Tables

**Figure 1 ijms-22-10239-f001:**
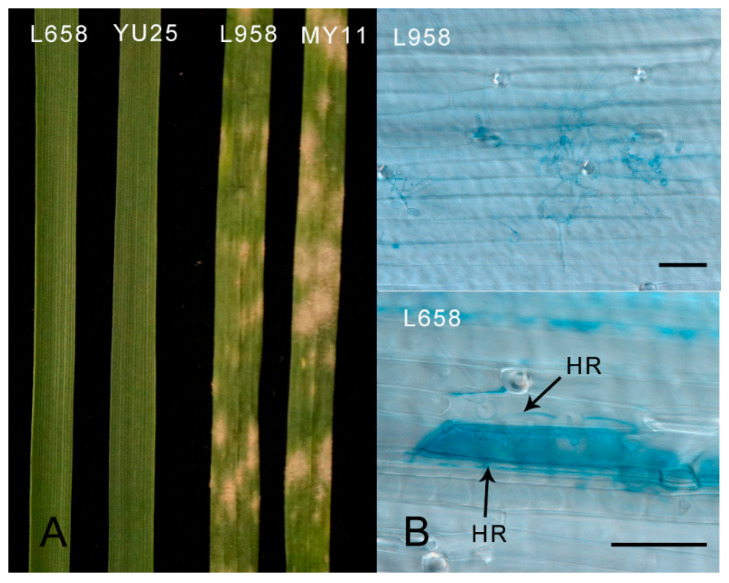
Resistance evaluations and cytological observations of cell death. (**A**) Resistance evaluations after 14 days of inoculation with *Bgt* at the seedling stage in wheat lines L658, YU25, L958, and MY11. (**B**) Hypersensitive response (HR) observed in wheat line L658 carrying *Pm40* and wheat line L958 lacking *Pm40* after 72 hpi. The dark bar indicates 50 μm.

**Figure 2 ijms-22-10239-f002:**
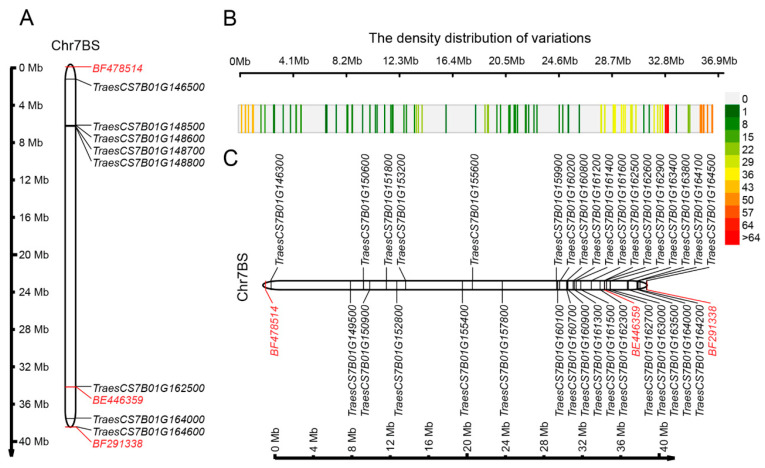
Distributions of NLR and RLK genes, identified variants, and the genes containing variations. (**A**) The physical locations of 7 RLK genes and 1 NLR gene in the *Pm40* mapping region. (**B**) Variants were distributed in the mapping interval of *Pm40*. The distribution of genes containing variants that may have an effect on the gene. (**C**) The distribution of genes that have variants that may affect the genes.

**Figure 3 ijms-22-10239-f003:**
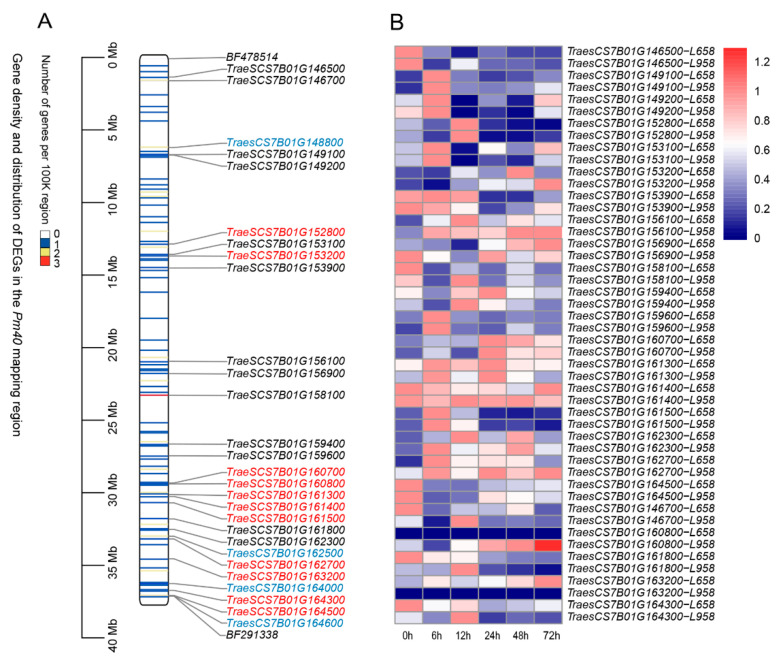
Distribution of differentially expressed genes and expression trends after inoculation with powdery mildew fungus. (**A**) Distribution of differentially expressed genes in the mapping interval of *Pm40* (the genes in red indicate those with mutations that probably affect genes). (**B**) The expression trends of DEGs in the resistant line L658 and susceptible sister line L958 at 0 (without inoculation), 6, 12, 24, 48, and 72 h post-inoculation. Heat map drawn based on log2 RPKM values +1 converted to percentages.

**Figure 4 ijms-22-10239-f004:**
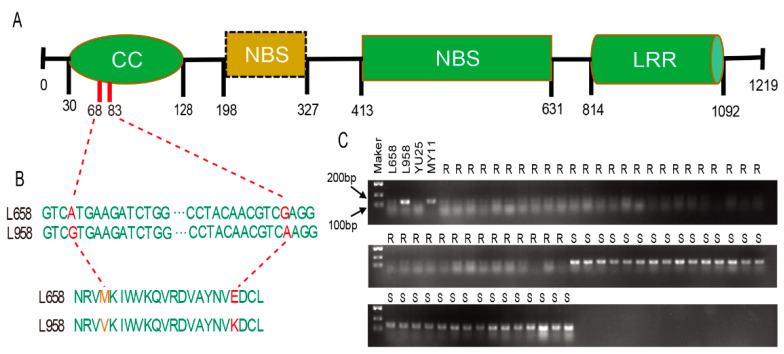
Sequence analysis of *TraesCS7B01G164000* and SNP identification in the L658 sister line. (**A**) The *TraesCS7B01G164000* domain composition and mutation sites (mutations in red indicate changes in amino acids, and vertical lines indicate positions of corresponding domains). (**B**) Two SNPs located in the coding region of *TraesCS7B01G164000* were found in the *Pm40*-carrying wheat line L658 and susceptible sister line L958. (**C**) SNP validation of *TraesCS7B01G164000* in 39 resistant lines and 30 susceptible sister lines of L658 by AS-PCR.

**Figure 5 ijms-22-10239-f005:**
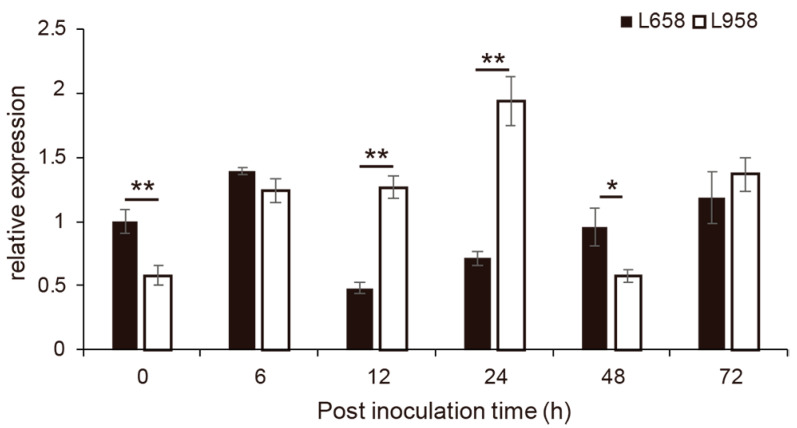
Relative expression changes of *TraesCS7B01G164000* in the resistant line L658 and susceptible sister line L958 inoculated with powdery mildew. Statistical significance was determined using an independent sample *t*-test. The asterisks represent significant differences as follows: ** *p* < 0.01 and * *p* < 0.05. The asterisk at the top of the bars represents the difference between L658 and L958 interactions at each same time point.

**Table 1 ijms-22-10239-t001:** NBS-LRR and RLK genes within the *Pm40* region were annotated in Pfam, SwissPort, Eggnog, and NR Database.

Gene ID	Pfam Annotation	Swissprot Annotation	Eggnog Annotation	NR Annotation
*TraesCS7B01G146500*	Protein tyrosine kinase	Serine/threonine-protein kinase	Signal transduction mechanisms	Serine/threonine-protein kinase
*TraesCS7B01G148500*	Protein tyrosine kinase	Wall-associated receptor kinase	Signal transduction mechanisms	Wall-associated receptor kinase
*TraesCS7B01G148600*	Protein tyrosine kinase	Wall-associated receptor kinase	Signal transduction mechanisms	Wall-associated receptor kinase
*TraesCS7B01G148700*	Protein tyrosine kinase	Wall-associated receptor kinase	Signal transduction mechanisms	Wall-associated receptor kinase
*TraesCS7B01G148800*	Protein tyrosine kinase	Wall-associated receptor kinase	Signal transduction mechanisms	Wall-associated receptor kinase
*TraesCS7B01G162500*	Protein tyrosine kinase	Chitin elicitor receptor kinase	Signal transduction mechanisms	predicted protein
*TraesCS7B01G164000*	NB-ARC domain	Putative disease resistance protein	Signal transduction mechanisms	resistance protein RGA2
*TraesCS7B01G164600*	Protein tyrosine kinase	PTI1-like tyrosine-protein kinase	Signal transduction mechanisms	PTI1-like tyrosine-protein kinase

**Table 2 ijms-22-10239-t002:** The effects of genetic variations in the Pm40 region predicted by mutation annotation.

Gene ID	Variants_Impact_HIGH				Variants_Impact_MODERATE
	Frameshift	Splice Acceptor	Splice Donor	Stop Gained	Missense Variant
*TraesCS7B01G146300*	0	0	0	0	3
*TraesCS7B01G149500*	0	3	0	0	2
*TraesCS7B01G150600*	0	0	6	0	2
*TraesCS7B01G150900*	2	0	0	0	0
*TraesCS7B01G151800*	0	0	0	0	3
*TraesCS7B01G152800*	0	0	0	2	0
*TraesCS7B01G153200*	0	0	0	0	1
*TraesCS7B01G155400*	0	0	0	0	3
*TraesCS7B01G155600*	0	0	0	0	1
*TraesCS7B01G157800*	0	0	0	0	2
*TraesCS7B01G159900*	0	0	0	0	1
*TraesCS7B01G160100*	0	0	0	0	2
*TraesCS7B01G160200*	0	0	0	0	2
*TraesCS7B01G160700*	0	0	0	0	1
*TraesCS7B01G160800*	1	0	0	0	3
*TraesCS7B01G160900*	0	0	0	0	1
*TraesCS7B01G161200*	0	0	0	0	5
*TraesCS7B01G161300*	0	0	0	0	4
*TraesCS7B01G161400*	1	0	0	0	2
*TraesCS7B01G161500*	0	0	0	0	2
*TraesCS7B01G161600*	0	0	0	0	2
*TraesCS7B01G162300*	2	0	0	0	1
*TraesCS7B01G162500*	0	0	0	0	3
*TraesCS7B01G162600*	0	0	0	0	1
*TraesCS7B01G162700*	0	0	0	0	5
*TraesCS7B01G162900*	0	0	0	0	1
*TraesCS7B01G163000*	0	0	0	0	1
*TraesCS7B01G163400*	0	0	0	1	0
*TraesCS7B01G163500*	0	0	0	0	1
*TraesCS7B01G163800*	0	0	0	0	2
*TraesCS7B01G164000*	0	0	0	0	2
*TraesCS7B01G164100*	0	1	0	0	0
*TraesCS7B01G164200*	0	0	0	0	1
*TraesCS7B01G164500*	0	0	0	0	7

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
