# Peer review of "Identification and Cloning of a CC-NBS-NBS-LRR Gene as a Candidate of Pm40 by Integrated Analysis of Both the Available Transcriptional Data and Published Linkage Mapping"

_ijms, 2021, doi:10.3390/ijms221910239_

Round 1
Reviewer 1 Report
1. were the gene annotation taken from published studies or also self computed using various mapping tools?
2.is BF478514 derived from a wheat gene? maybe no counterpart in the IWGSC annotation? or does it only include the RLK genes? (reg. Figure 2)
3. (reg 2.3) "Transcriptomic data showed that 100 out of 185 genes in the Pm40 region expressed 159 after inoculation with Bgt", what was the criteria whether it is expressed?
4. For me, table-2 is rather a supplement figure, expect you indicate gene functions; use different and more relevant labels for "Variants_impact_HIGH Variants_impact_MODERATE "
5. the 24 DEGs, are these sign. enriched in these mentioned function categories?
6. figure 3. y-axis label should be added
minor:
typo: p7. modifier--impact
Author Response
Response to Reviewer 1 Comments
Point 1: were the gene annotation taken from published studies or also self computed using various mapping tools?
Response 1: For the annotations of genes in the Pm40 chromosome region, we used the gene annotations of the Chinese spring wheat reference genome (IWGSC RefSeq v1.0-https://wheat-urgi.versailles.inra.fr/) and further updated the functional annotations with blast2go software to obtain the latest data information.
Point 2: is BF478514 derived from a wheat gene? maybe no counterpart in the IWGSC annotation? or does it only include the RLK genes? (reg. Figure 2)
Response 2: The BF478514 is an EST-STS marker derived from gene expression sequence of wheat, which completely overlapped with the wheat gene TraesCS7B01G146200, so accurate gene annotations can be obtained.
Point 3: (reg 2.3) "Transcriptomic data showed that 100 out of 185 genes in the Pm40 region expressed after inoculation with Bgt", what was the criteria whether it is expressed?
Response 3: To filter low-expression genes, the following criteria were used: expression level of genes CPM(count per million)value > 1, and sum of CPM > 3 in all samples. We have added this part into materials and methods of the manuscripts.
Point 4: For me, table-2 is rather a supplement figure, expect you indicate gene functions; use different and more relevant labels for "Variants_impact_HIGH Variants_impact_MODERATE "
Response 4: Thank you very much for the valuable comment. Table 2 not only contains the information of the variations, but also contains the effects of genetic variations in the Pm40 region, which is consistent with the mutational sites detected by the cloned gene sequences. Therefore, we think it is very important to keep them in the text.
Point 5: the 24 DEGs, are these sign. enriched in these mentioned function categories?
Response 5: We only briefly listed the functional annotations of the 24 DEGs, because the functions of these genes have no obvious relationship with disease resistance. And table S2 contains the functional annotations of all 24 DEGs.
Point 6: figure 3. y-axis label should be added
Response 6: Thank you very much for pointing out this problem. We have added y-axis label for Figure 3.
Point 7: typo: p7. modifier—impact
Response 7: We have further proofread the Professional Vocabulary, please refer to it (http://snpeff.sourceforge.net).
Reviewer 2 Report
The manuscript results look robust, also the paper is very well presented the data and discuss the results. This result will confirm that Pm40 gene is still valid against powdery mildew and also support the plant breeder to continue to use this gene against powdery mildew in barley breeding programs.
For the novelty of this manuscript, there is more than one study describing this gene so I will say it's not a novel study (Luo PG, Luo HY, Chang ZJ, Zhang HY, Zhang M, Ren ZL. Characterization and chromosomal location of Pm40 in common wheat: a new gene for resistance to powdery mildew derived from Elytrigia intermedium. Theor Appl Genet. 2009 Apr;118(6):1059-64. doi: 10.1007/s00122-009-0962-0. Epub 2009 Feb 5. PMID: 19194691.)
- For the significance of content, The study is well done and covers every corner of the subject. -Quality of presentation: The manuscript is very well presented. -Scientific soundness: The manuscript has followed the scientific way of describing and introduce the problem and solve it. -Interest to the readers and overall merit: For this point, most plant breeders of wheat will be interested in studies like this.I recommend publishing the manuscript in present form.
Author Response
Thank you very much for your positive comments, and we have further proofread the manuscript carefully.
Round 2
Reviewer 1 Report
remarks were properly addressed